

# Change in mental health, physical health, and social relationships during highly restrictive lockdown in the COVID-19 pandemic: evidence from Australia

Shane L. Rogers[1] and Travis Cruickshank[2]

[1] School of Arts and Humanities, Edith Cowan University, Joondalup, Western Australia, Australia
[2] School of Medical and Health Sciences, Edith Cowan University, Joondalup, Western Australia, Australia

## ABSTRACT

**Background**. A novel coronavirus first reported in Wuhan City in China in 2019 (COVID-19) developed into a global pandemic throughout 2020. Many countries around the world implemented strict social distancing policies to curb the spread of the virus. In this study we aimed to examine potential change in mental/physical health and social relationships during a highly restrictive COVID-19 lockdown period in Australia during April 2020.

**Methods**. Our survey ($n = 1,599$) included questions about concerns, social behaviour, perceived change in relationship quality, social media use, frequency of exercise, physical health, and mental health during COVID-19 lockdown (April, 2020).

**Results**. When estimating their mental health for the previous year 13% of participants reported more negative than positive emotion, whereas this increased to 41% when participants reflected on their time during COVID-19 lockdown. A substantial proportion (39–54%) of participants reported deterioration in mental health, physical health, financial situation, and work productivity. However, most of these participants reported 'somewhat' rather than 'a lot' of deterioration, and many others reported 'no change' (40–50%) or even 'improvement' (6–17%). Even less impact was apparent for social relationships (68% reported 'no change') as participants compensated for decreased face-to-face interaction via increased technology-mediated interaction.

**Conclusions** . The psychological toll of COVID-19 on Australians may not have been as large as other parts of the world with greater infection rates. Our findings highlight how technology-mediated communication can allow people to adequately maintain social relationships during an extreme lockdown event.

# INTRODUCTION

The year 2020 was dominated by the influence of the novel coronavirus COVID-19 pandemic (*Gruber et al., 2020*; *Hsiang et al., 2020*). The disruption caused directly by virus containment policies, and the flow on effects of economic decline, were a source of stress

Corresponding author
Shane L. Rogers,
shane.rogers@ecu.edu.au

impacting populations around the world (*Gruber et al., 2020*; *Holmes et al., 2020*). The present study adds to the literature examining the mental health impacts of the COVID-19 pandemic by focusing on the experience of people in Australia. We report the findings of a survey undertaken in April 2020 when the Australian government had implemented very strict social distancing policies to contain the virus. We focus on quantifying the *emotional well-being* of participants during lockdown, and the impact on *social relationships* and *communication patterns*. While studies from around the world have consistently reported broad negative impacts of lockdowns on mental health, it is less understood how different types of social relationships might have been impacted by lockdown experiences.

## The impact of COVID-19 lockdown on mental health around the world, and in Australia

Government policies enforcing restrictive lockdown to curb the spread of COVID-19 have the potential to negatively impact on fundamental psychological needs. According to self-determination theory (SDT) three core psychological needs are competence, autonomy and relatedness (*Ryan & Deci, 2000*). Mandated social distancing during a pandemic may detract from an individual's sense of autonomy (i.e., sense of personal control and freedom) and relatedness (i.e., sense of closeness to other people). The flow on negative impact on the economy may further exacerbate stress as people feel a threat to their competence (i.e., sense of self-efficacy) as jobs are threatened or lost. Similarly, the same kind of logic can be applied to Maslow's hierarchy of needs that includes physiological, safety, social, esteem, and self-actualisation elements (for a discussion see: *Ryan et al., 2020*).

Quarantine (lockdown) during prior pandemics such as the severe acute respiratory syndrome (SARS) and Middle East respiratory syndrome (MERS) were reported to negatively affect psychological well-being (*Brooks et al., 2020*). Research on people's experience during COVID-19 lockdowns around the world have produced consistent findings of increased levels of stress, anxiety and depression (*Baloch et al., 2021*; *Benke et al., 2020*; *Bruno et al., 2021*; *Ebrahimi, Hoffart & Johnson, 2020*; *Fiorenzato et al., 2020*; *Gao et al., 2020*; *Hamadani et al., 2020*; *Holingue et al., 2020*; *Holman et al., 2020*; *Huang & Zhao, 2020*; *Kalaitzaki, 2020*; *Lee, 2020*; *Marashi et al., 2020*; *Mazza et al., 2020*; *Ozamiz-Etxebarria et al., 2020*; *Patrick et al., 2020*; *Pierce et al., 2020*; *Qiu et al., 2020*; *Roy et al., 2020*; *Sameer et al., 2020*; *Saraswathi et al., 2020*; *Twenge & Joiner, 2020*; *Wang et al., 2020*; *Zacher & Rudolph, 2020*).

Factors reported as being associated with higher levels of emotional distress during lockdown are being younger, female, having lower education, having pre-existing medical conditions, increased social media use, rumination on COVID-19 (*Baloch et al., 2021*; *Benke et al., 2020*; *Bruno et al., 2021*; *Bu et al., 2021*; *Ebrahimi, Hoffart & Johnson, 2020*; *Fiorenzato et al., 2020*; *Gao et al., 2020*; *Holman et al., 2020*; *Hsiang et al., 2020*; *Huang & Zhao, 2020*; *Kalaitzaki, 2020*; *Lee, 2020*; *Mazza et al., 2020*; *Ozamiz-Etxebarria et al., 2020*; *Qiu et al., 2020*; *Roy et al., 2020*; *Wang et al., 2020*), whereas keeping physically and socially active have been reported as protective factors (*Brand, Timme & Nosrat, 2020*; *Bu et al., 2021*; *Ebrahimi, Hoffart & Johnson, 2020*; *Galle et al., 2020*; *Marashi et al., 2020*).

Some Australian surveys have been conducted showing that the lockdown experience, similar to other parts of the world, had a broadly negative impact on mental health (*Biddle et al., 2020a*; *Brand, Timme & Nosrat, 2020*; *Fisher et al., 2020*; *Li et al., 2020*; *Newby et al., 2020*; *Phillipou et al., 2020*; *Rossell et al., 2021*; *Stanton et al., 2020*; *Titov et al., 2020*; *van Agteren et al., 2020*; *Westrupp et al., 2021*). In a survey of 5,070 Australians in the first week of April 2020, *Newby et al. (2020)* found that 55% of respondents felt their mental health had worsened a little, and 23% a lot. Primary worries underlying this were that 50% or more indicated worrying about catching the virus, feeling moderately-extremely lonely, and feeling uncertain about their financial situation. In a survey with 13,829 respondents, *Fisher et al. (2020)* also found overall elevated levels of psychological distress.

## The potential impact of COVID-19 lockdown on social relationships

One underlying psychological mechanism to explain how COVID-19 lockdown might reduce mental health is via negative impacts on social relationships. Social relationships require maintenance where the extent of required maintenance can vary a great deal depending on the closeness of the relationship, and personal qualities and preferences (*Blieszner & Ogletree, 2017*; *Fehr, 2004*; *Mesch & Talmund, 2006*; *Ogolsky & Bowers, 2012*). COVID-19 lockdown experiences have the potential to significantly impair core aspects of relationship maintenance by disrupting the frequency and quality of shared experiences, and interpersonal communication patterns. This could be associated with psychological consequences such as a decreased sense of closeness and a reduction in perceived relationship quality (*Aleman & Sommer, 2020*; *Lardone et al., 2020*).

Engaging in shared activities/experiences has been identified as one key aspect of relationship maintenance (*Daniels, Watson & Gedikli, 2017*; *Girme, Overall & Faingataa, 2014*; *Jolly et al., 2019*; *Mesch & Talmund, 2006*; *Rossignac-Milon & Higgins, 2018*). The closure of sources of social activities (e.g., restaurants, bars, sporting clubs, gyms, movie theatres, etc.) during COVID-19 lockdown arguably has the potential to impair social relationships via disrupting engagement in shared activities for relationship maintenance. While engaging in shared activities can facilitate social bonds, so too can the simple act of conversing with others (*Rossignac-Milon & Higgins, 2018*). Communicative acts such as self-disclosure (*Sprecher & Treger, 2015*), reminiscing (*Roberts, 2018*), sharing news (*Reis et al., 2010*) and gossip (*Yucel et al., 2020*), all have the potential to serve as bonding experiences. Therefore, social distancing policies associated with COVID-19 lockdown has potential to reduce the amount or frequency of social interaction, as people spend more time at home.

Lockdown also has the potential to impact upon the type of social interaction, with an increased amount of communication via technology (e.g., phone, email, social media) to compensate for reduced opportunities for face-to-face interaction. Research suggests that communication via technology can be considered as impoverished compared to face-to-face communication via a reduction in certain verbal (e.g., tone of voice) or non-verbal (e.g., eye contact, gestures, or body posture) communicative cues (*Colvin et al., 2004*; *Kock, 2002*; *Kock, 2004*; *Lieberman & Schroeder, 2020*; *Walther, 1996*; *Walther & Parks, 2002*). As a consequence it is not surprising that people generally have a preference for face-to-face

communication over technology-mediated communication in most instances (*Flaherty, Pearce & Rubin, 1998*; *Wilson et al., 2020*). Therefore, even if a person tries to compensate for reduced face-to-face interaction via a corresponding increase in technology-mediated communication, theoretically there remains potential for a negative impact on perceived relationship quality if the subsequent technology-mediated communication is experienced as a lesser form of interaction.

## Research investigating social relationships during COVID-19 lockdown

As discussed in the prior section, it is reasonable to expect that social relationships might be negatively impacted during COVID-19 lockdown. However, recent studies that have investigated loneliness during the pandemic have been reporting mixed findings. Some studies have reported an increase in loneliness (*Bu, Steptoe & Fancourt, 2020*; *Killgore et al., 2020*; *Van Tilburg et al., 2020*), whereas others have reported there has been no significant increase in loneliness during lockdowns compared to pre-pandemic levels (*Groarke et al., 2020*; *Luchetti et al., 2020*).

Few studies have directly asked targeted questions about perceived relationship quality during the pandemic (*Biddle et al., 2020b*; *Bulow et al., 2020*; *Cooper, Pauletti & Di Donato, 2020*; *Philpot et al., 2021*), and even less about the use of technology-mediated communication (*Monin et al., 2020*). In a recent Australian study *Biddle et al. (2020b)* asked their 3,249 respondents from the general public "How has the quality of your relationships with other people/family members in your household changed since the spread of COVID-19?". Responses were: 2% a lot more difficult/strained, 15% a little more difficult/strained, 55% no change, 23% a little closer/stronger, and 5% a lot closer/stronger. *Biddle et al.*'s (*2020b*) study provides some evidence to suggest that during the lockdown in April there may not have been any broad large negative impact on social relationships in the Australian population.

## The present study

Our study aims to contribute to the emerging literature on the mental and physical health impacts of government mandated lockdown policies on the public during COVID-19. We also take an in-depth look at how communication patterns changed regarding face-to-face and technology-mediated communication to examine how these different modes of communication are associated with self-reported quality of social relationships. Our survey was conducted from 14th–30th April in 2020 when all non-essential businesses were closed, and people were not supposed to congregate in groups larger than two people outside of their immediate household members. We were anticipating that like other Australian studies investigating the same period in time, we would find a substantial portion of participants self-reporting deterioration across their mental health, physical health, and financial situation (*Fisher et al., 2020*; *Newby et al., 2020*; *Phillipou et al., 2020*).

Based on background psychological literature on relationship maintenance, and the impoverished nature of technology mediated communication (when compared to face-to-face communication), we were initially predicting that COVID-19 lockdown would be

associated with a negative impact on social relationships in our sample. However, based on *Biddle et al. (2020b)* results (that was published after our study commenced), we were subsequently anticipating that social relationships might not be greatly impacted for most individuals. Extending upon *Biddle et al. (2020b)*, we further explore potential changes in participant behaviour regarding the extent of face-to-face and technology-mediated social interaction, and whether (or not) changes in social interaction are associated with changes in perceived relationship quality.

We also asked participants about a range of concerns they were experiencing about COVID-19 to compare how concerns about their personal relationships compares with other concerns. If we find that social relationships are not being heavily impacted, then we also expect that self-reported concerns about social relationships will overall be lower compared to other concerns. Prior research has indicated that concerns regarding catching the virus and the economy have typically been high during the pandemic (*Ebrahimi, Hoffart & Johnson, 2020*; *Newby et al., 2020*; *Saraswathi et al., 2020*; *Timming, French & Mortensen, 2021*; *Wang et al., 2020*). We also look at other factors that have been examined in other parts of the world but not yet given much attention by Australian researchers (i.e., frequency of exercise, COVID-19 rumination, and social media exposure). Based on what has recently been reported in the literature, we hypothesize that in our sample higher frequency of exercise will be positively associated with emotional well-being (*Bu et al., 2021*; *Ebrahimi, Hoffart & Johnson, 2020*; *Galle et al., 2020*), whereas increased rumination on COVID-19 and increased use of social media will be negatively associated with emotional well-being (*Bu et al., 2021*; *Gao et al., 2020*; *Hsiang et al., 2020*; *Lee, 2020*; *Roy et al., 2020*).

## MATERIALS & METHODS

### Participants and procedure

An online survey was disseminated between 14th-30th April 2020 via social media channels (primarily Facebook). We received 1599 responses, 78% female. Age of participants in years: 18–21 (6%), 22–25 (4%), 26–30 (11%), 31–35 (13%), 36–40 (11%), 41–50 (22%), 51–60 (18%), 61–70 (11%), 71+ (3%). Employment status: Full-time (44%), part-time (19%), casual (12%), stay at home parent (8%), retired (9%), unemployed (8%). The industry worked in: Health care and social assistance (24%), Education and training (22%), Professional, scientific and technical services (9%), retail trade (6%), administration (5%), with the remaining 44% spread across other industries in smaller proportions. Most of our participants resided in Western Australia (81%), with the remainder of the sample coming from the various other Australian states and territories. Most participants (75%) were in a committed relationship at the time of the study. The lockdown period began from the start of April 2020, where gatherings were restricted to two people (except for immediate members of households). Participants in our sample had been in lockdown for an average of 18.84 days ($SD = 4.62$, $min = 14$, $max = 30$). Prior to running the survey, ethics approval was obtained from Edith Cowan University (Ref: 2020-01305-ROGERS). Informed consent was obtained from a dedicated question embedded at the beginning of the survey asking participants to consent for their data to be used for research purposes.

## Survey emasures
### Emotional and physical well-being

We measured emotional and physical well-being via the Brief Emotional Experience Scale (BEES) and Brief Emotional Experience Physical Scale (BEEPS) (*Rogers, Barblett & Robinson, 2016*; *Rogers, Cruickshank & Nosaka, 2021*; *Skead & Rogers, 2016*; *Skead, Rogers & Doraisamy, 2018*). For the BEES, participants rate how they are emotionally feeling for the adjectives Happy, Worried, Calm, Sad, Confident, and Afraid, on a response scale: (1) Not at all (2) A little bit (3) A lot. For the BEEPS, participants to rate how they are physically feeling for the adjectives Healthy, Lethargic, Strong, Unfit, Energetic, Weak, using same response scale as the BEES. For both measures an overall score can be created by averaging across the three positive and negative adjectives separately, and then subtracting the negative from the positive score. This provides an overall score that can range from +3 to -3 where a score above zero indicates greater positive than negative emotion, and a score below zero greater negative than positive emotion. The BEES and BEEPS were repeated twice in the survey. Participants answered by reflecting on their *past month*, and then answered a second time reflecting on the *same time last year*. In the present study Cronbach Alpha values for the BEES and BEEPs across the four measurements ranged from .85 to .87.

### Exercise

We measured frequency of exercise via questions previously used by *Skead & Rogers (2016)* and *Skead, Rogers & Doraisamy (2018)*. Participants rated how often they had been engaging in light (e.g., walking), moderate (e.g., jogging), and high (e.g., running) intensity exercise, on a response scale: (0) Never (1) About once a week or less (2) A few times per week (3) About once a day. These questions were asked twice, by asking participants to reflect on the *past month*, and then also on the *same time last year*. As per *Skead & Rogers (2016)* we created a composite exercise variable giving greater weightings to moderate and high intensity exercise via the formula: Low + (2*moderate) + (3*high). This provides a frequency of exercise score that ranges from a minimum of zero to maximum of 18.

### Perceived impact of COVID-19 lockdown on different aspects of life, and concerns about COVID-19

Participants were asked how COVID-19 lockdown may have impacted upon their mental health, physical health, financial situation, and social relationships over the past month on a response scale: (1) Deteriorated a lot (2) Deteriorated somewhat (3) no change (4) Improved somewhat (5) Improved a lot. Participants were also asked how concerned they were about several COVID-19 related aspects, such as the ability to purchase necessities and impact on the economy, on a response scale: (1) Not concerned (2) Slightly concerned (3) Very concerned (4) Extremely concerned. The full list of concerns is shown in results sub-section 'Concerns related to COVID-19'.

### Attention given to COVID-19

We assessed participant attention given to COVID-19 over the past month via four items that asked how much they had been thinking about COVID-19, talking with others about

COVID-19, following information via news stories, and via more official sources. These questions were rated on a response scale: (1) About once a week or less; (2) A few times per week; (3) About once a day; (4) Multiple times per day.

### Self-perceived social behaviour change due to COVID-19

We asked participants to reflect on a series of questions regarding their social behaviour change due to COVID-19. The first three items asked about how much they had been distancing from others, isolating from others, and spending time on social media. The remainder of items asked the extent to which they had been interacting with their partner, friends, family, or work colleagues either face-to-face, or via technologically mediated modes of communication (i.e., phone, email, online). All these items used a scale with the following response items: (1) A lot less; (2) Somewhat less; (3) No change; (4) Somewhat more; (5) A lot more.

## RESULTS

Throughout our results we use Pearson $r$ as a measure of effect size for mean comparison tests, in addition to also reporting it for standard correlation analyses (*Field, 2018*). Our interpretation of the magnitude of $r$ values follows broad guidelines of small (0.1), medium (0.3), and large (0.5) provided by *Cohen (1992)*.

### Perceived impact of COVID-19 lockdown on emotional well-being, physical well-being, relationships, and work

We asked participants a series of questions about their emotional well-being, physical well-being, and exercise reflecting over the prior month, and for the same time last year. The results from these measures are presented in Fig. 1. Both emotional ($t$ (1598) = 24.45, $p < .001$, $r = .52$) and physical ($t$ (1598) = 19.58, $p < .001$, $r = .44$) well-being were found to be lower during the COVID-19 lockdown period compared to participant estimates of their health the same time last year. More specifically, while 13% of participants estimated greater negative than positive emotion for the prior year (i.e., BEES score <0), this increased to 41% of participants for the prior month. Similarly, 19% of participants reported greater negative than positive physical well-being for the prior year (i.e., BEEPS score <0), increasing to 42% for the prior month. As indicated by the effect sizes ($r$), and as can be seen in Fig. 1, these represent statistically moderate effects. When comparing estimates of last year with the present month, we also found an overall decrease in self-reported frequency of exercise ($t$ (1598) = 9.80, $p < .001$, $r = .24$). However as indicated by the effect size ($r$), and can be seen in Fig. 1, this represents a relatively small effect.

Correlations among self-reported emotional well-being, physical well-being, and exercise, during lockdown and last year, are presented in Table 1. Emotional well-being (BEES) was found to be positively associated with physical well-being (BEEPS) during lockdown (*Pearson r = .60*, $p < .001$), and last year (*Pearson r = .57*, $p < .001$). Only weak positive associations between current month and last year values were found for both the emotional wellbeing and physical wellbeing suggesting that participants were able to dissociate last year from the current year when making their appraisals. Frequency

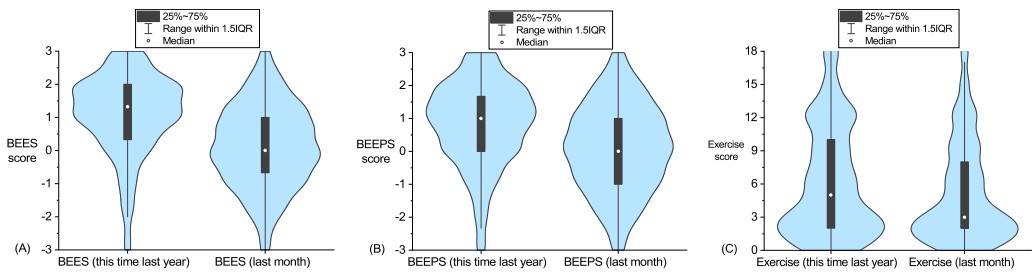

**Figure 1 Emotional well-being, physical well-being, and frequency of exercise during lockdown compared with estimates for same time last year.** Violin plots of participant estimates for last year compared to the prior month during COVID-19 lock-down for (A) emotional well-being as measured by the Brief Emotional Experience Scale (BEES), (B) physical well-being as measured by the Brief Emotional Experience Physical Scale, (C) exercise as measured by three items asking frequency of low, moderate and high intensity exercise combined into an overall composite frequency of exercise score.

**Table 1 Pearson correlations between self-report ratings for the BEES, BEEPS, and exercise during lockdown, and reflections on the prior year.**

|  | 1. | 2. | 3. | 4. | 5. | 6. |
|---|---|---|---|---|---|---|
| 1. BEES (lockdown) | 1 |  |  |  |  |  |
| 2. BEEPS (lockdown) | .60* | 1 |  |  |  |  |
| 3. Exercise (lockdown) | .17* | .40* | 1 |  |  |  |
| 4. BEES (last year) | .18* | .20* | .01 | 1 |  |  |
| 5. BEEPS (last year) | .10* | .34* | .18* | .57* | 1 |  |
| 6. Exercise (last year) | .01 | .12* | .56* | .12* | .48* | 1 |

**Notes.**
*$p < .001$.

of exercise was found to positively associate with physical well-being, during lockdown (*Pearson r = .40, p < .001*), and last year (*Pearson r = .48, p < .001*). Whereas there were only weak positive associations between frequency of exercise and emotional well-being. This replicates prior work by *Skead & Rogers (2016)* that suggests exercise may have an indirect benefit for mental well-being via enhancing one's sense of physical well-being.

## Perceived impact of COVID-19 lockdown on emotional well-being, physical well-being, relationships, and work

We directly asked participants to estimate the impact of COVID-19 lockdown regarding their mental health, physical health, finances, work, and relationships, see Fig. 2. A substantial proportion of participants indicated at least some deterioration of their mental health (54%), physical health (41%), financial situation (39%), and productivity with work (41%). Note that for most individuals reporting a deterioration, the response is *somewhat* rather than *a lot* (see Fig. 2). Also, except for mental health, the most common response for physical health, financial situation, and work productivity was *no change*. Interestingly, there appears less impact on social relationships, as only approximately 17% indicated deterioration across different types of relationships, while approximately 66–72% reported no change, and 8–16% reported improvement. As will be shown later in the results section,
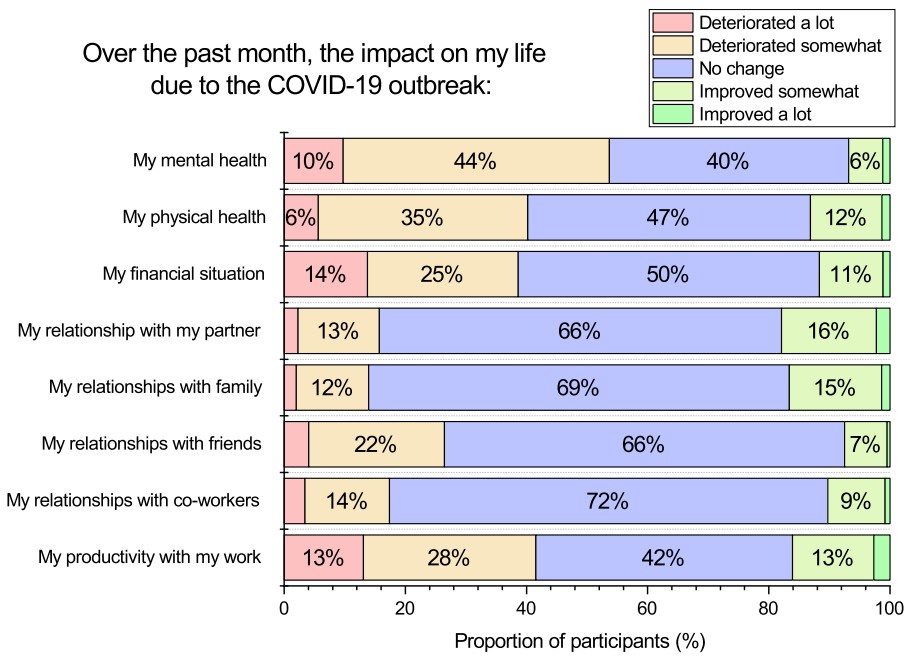

**Figure 2  Participants ratings of the direct impact they felt COVID-19 lockdown was having upon multiple aspects of their lives.**

many participants appeared to compensate for a decrease in face-to-face contact using technology to keep in contact with others.

Inter-correlations among the variables in Fig. 2 are presented in Table 2. Participants' present month emotional well-being score was positively associated with participants' self-rating of COVID-19 lockdown impact on their mental health ($Spearman\ r = .64, p < .001$), and present month physical well-being was positively associated with their rating of COVID-19 lockdown impact on their physical health ($Spearman\ r = .49, p < .001$). There are only weak positive associations among most other variables.

## Concerns related to COVID-19

We asked participants the extent they were concerned about a range of social and economic issues due to COVID-19, see Fig. 3. We found a substantial proportion of participants to have concern across all issues, however the two stand outs are concern about friends and family catching COVID-19 (63% very/extremely concerned), and a negative impact on the economy (73% very/extremely concerned). Note that concern over personal relationships suffering is much lower in comparison (20% very/extremely concerned).

Inter-correlations among the different concerns, and relationships with emotional well-being measures, are presented in Table 3. All concerns were found to be negatively associated with emotional well-being. Associations among concerns varied depending on the similarity of the concerns. As an example, a strong association was found between self-concern catching COVID-19 and concern for friends/family catching COVID-19

**Table 2** Spearman correlations between mental/physical health, change in finances, change in work productivity, and change in relationship quality across partner, family, friends, and work colleagues.

|  | 1 | 2 | 3 | 4 | 5 | 6 | 7 | 8 | 9 | 10 |
|---|---|---|---|---|---|---|---|---|---|---|
| 1. BEES (lockdown) | 1 | | | | | | | | | |
| 2. BEEPS (lockdown) | .58* | 1 | | | | | | | | |
| 3. Mental health change | .64* | .46* | 1 | | | | | | | |
| 4. Physical health change | .35* | .49* | .39* | 1 | | | | | | |
| 5. Finances change | .26* | .14* | .22* | .17* | 1 | | | | | |
| 6. Relationship - partner | .24* | .18* | .27* | .20* | .11* | 1 | | | | |
| 7. Relationships - Family | .15* | .11* | .20* | .15* | .12* | .37* | 1 | | | |
| 8. Relationships - Friends | .20* | .19* | .23* | .18* | .10* | .20* | .34* | 1 | | |
| 9. Relationships - Work colleagues | .17* | .08 | .15* | .14* | .13* | .15* | .15* | .31* | 1 | |
| 10. Work productivity change | .22* | .21* | .28* | .19* | .24* | .14* | .10 | .13* | .30* | 1 |

**Notes.**

*$p < .001$.

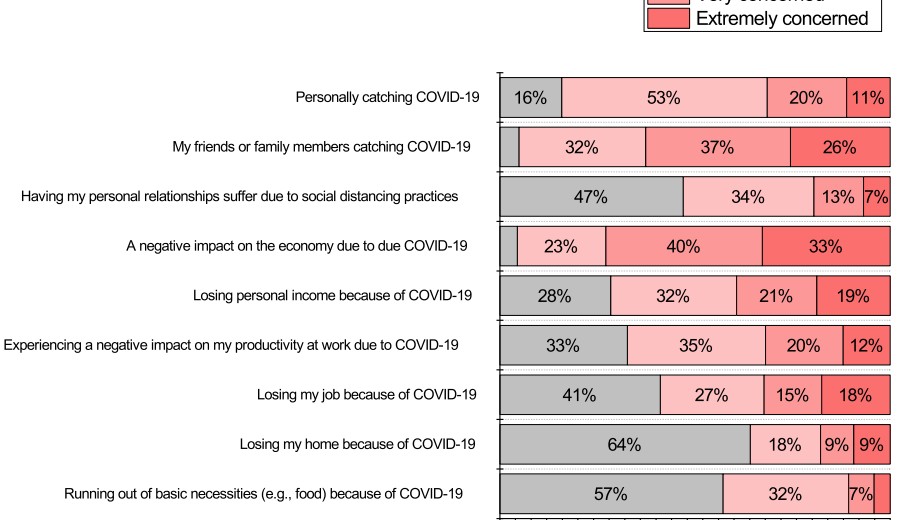

**Figure 3** Participant self-reported extent of COVID-19 related concern across multiple factors.

(*Spearman r = .60, p < .001*). Whereas these concerns about catching the virus were only weakly positively associated with all other concerns.

## Amount of personal attention given to COVID-19

We expected that participants would report giving a lot of personal attention to COVID-19 considering the extent of concern associated with it. Indeed, as can be seen in Fig. 4, between

**Table 3** Spearman correlations between mental health, and a range of concerns regarding COVID-19.

|  | 1 | 2 | 3 | 4 | 5 | 6 | 7 | 8 | 9 | 10 | 11 |
|---|---|---|---|---|---|---|---|---|---|---|---|
| 1.BEES (lockdown) | 1 | | | | | | | | | | |
| 2.Mental health change | .64* | 1 | | | | | | | | | |
| 3.Catching COVID (self) | −.27* | −.11* | 1 | | | | | | | | |
| 4.Catching COVID (F/F) | −.32* | −.17* | .60* | 1 | | | | | | | |
| 5.Relationships suffer | −.43* | −.42* | .10* | .17* | 1 | | | | | | |
| 6.Economy | −.18* | −.15* | .10* | .13* | .23* | 1 | | | | | |
| 7.Personal income | −.30* | −.22* | .14* | .24* | .24* | .26* | 1 | | | | |
| 8.Work productivity | −.32* | −.25* | .19* | .20* | .29* | .21* | .53* | 1 | | | |
| 9.Losing job | −.30* | −.22* | .15* | .19* | .26* | .22* | .68* | .52* | 1 | | |
| 10.Losing home | −.27* | −.17* | .16* | .19* | .31* | .20* | .53* | .36* | .55* | 1 | |
| 11.Running out of necessities | −.29* | −.21* | .28* | .26* | .35* | .10* | .24* | .27* | .28* | .44* | 1 |

Notes.

*$p < .001$.

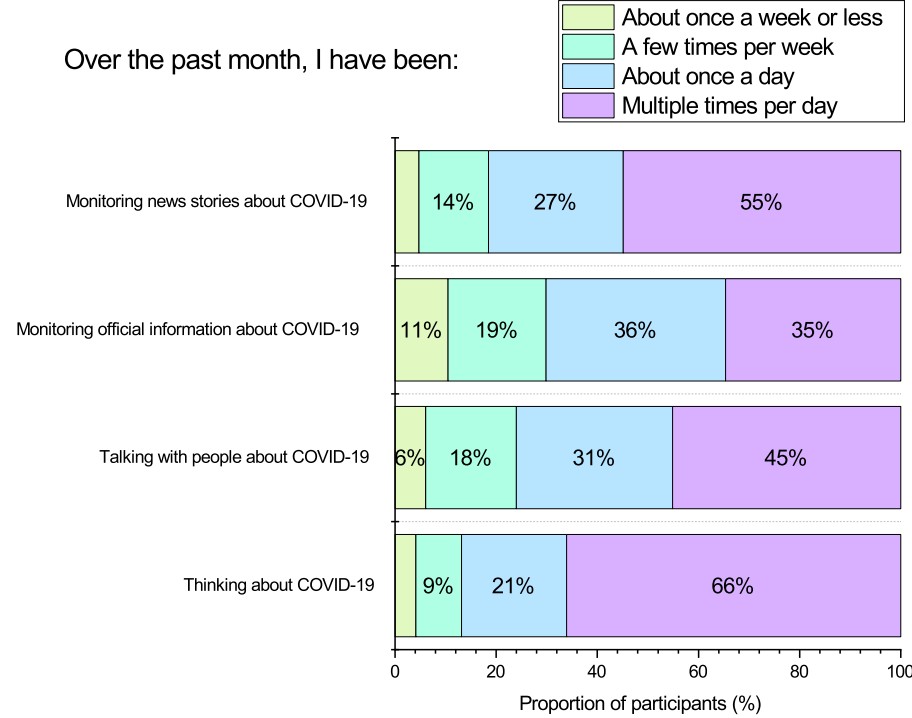

**Figure 4** Participant self-reported frequency of engagement with COVID-19 information, talking with people about COVID-19, and thinking about COVID-19.

35–66% of participants were thinking about COVID-19, monitoring news information, and talking with others about COVID-19 *multiple times per day*.

Inter-correlations among items are presented in Table 4. Across the items the clearest negative association with emotional well-being was the item 'thinking about COVID-19'

**Table 4** Spearman correlations between mental health, and amount of attention given to COVID-19.

|  | 1. | 2. | 3. | 4. | 5. | 6. |
|---|---|---|---|---|---|---|
| 1.BEES (lockdown) | 1 |  |  |  |  |  |
| 2.Mental health change | .64* | 1 |  |  |  |  |
| 3.Monitoring news | −.12* | −.06 | 1 |  |  |  |
| 4.Monitoring official sources | −.11* | −.03 | .64* | 1 |  |  |
| 5.Talking about COVID | −.10* | −.07 | .42* | .40* | 1 |  |
| 6.Thinking about COVID | −.27* | −.18* | .46* | .40* | .55* | 1 |

**Notes.**
*p < .001.

(*Spearman r* = −.27, *p* < .001). Therefore, we found a moderate association between rumination about COVID-19 and lower emotional well-being.

## Perceived changes in social behaviour due to COVID-19

Many participants reported a substantial shift in their social behaviour in response to the social distancing policies in place at the time of the survey, see Fig. 5. Many participants reported more distancing from others (88%), isolating themselves from other people (75%), and spending more time on social media (63%). Spending more time on social media was found to be negatively associated with BEES emotional well-being (*Spearman r* = -.28, *p* < .001), and perceived change in mental health (*Spearman r* = -.25, *p* < .001). Participant's face-to-face interaction time typically increased for time with one's partner (52%), and decreased for time with family (54%), friends (86%), and work colleagues (64%). Whereas technology mediated interaction time typically increased for family (57%), friends (56%), and work colleagues (49%). Therefore, results suggest that people were compensating for the decreased face-to-face interaction time with increased technology mediated interaction time.

## Associations between change in communication and change in relationship quality

Correlations between perceived change in relationship quality and change in the extent of both face-to-face and technology-mediated communication are presented in Table 5. A consistent weak-moderate positive association between change in face-to-face interaction and change in each type of relationship was found. Weak-moderate positive relationships were also found between change in technology mediated communication and perceived change in quality for relationships with family, friends, and work colleagues (but not partner).

There is a high proportion of participants reporting *no change* in perceived relationship quality across all relationship types. Therefore, the relatively low correlations in Table 5 might represent an under-representation of the strength of association between the variables due to a lack of spread across the response options, and/or non-linear patterns of association. In Fig. 6 we graphically present the associations between the variables which helps to provide a clearer indication of how both face-to-face and technology-mediated communication are associated with perceived change in relationship quality. For example,

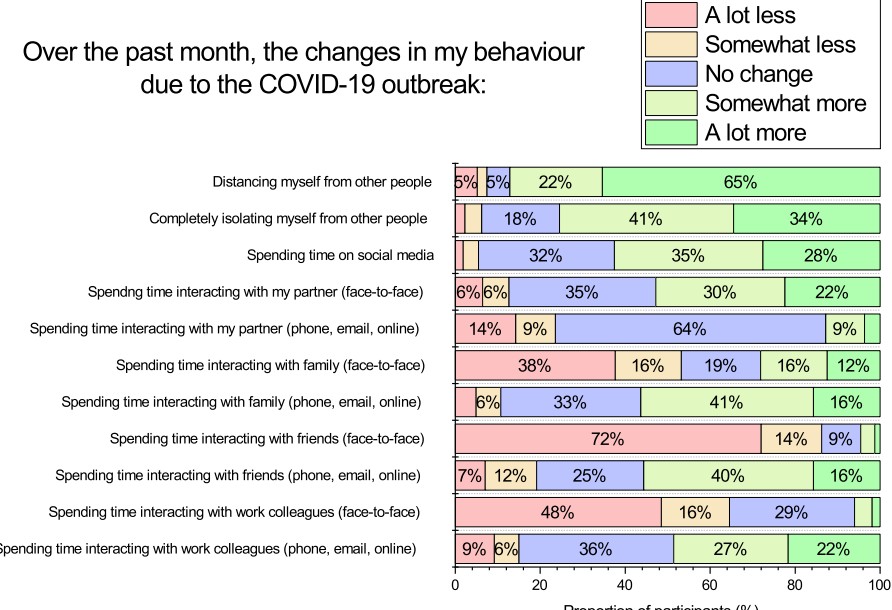

**Figure 5** Participant self-reported changes in social behaviour during the COVID-19 lock-down period.

**Table 5** Spearman correlations between change in perceived relationship quality with change in face-to-face and technology-mediated interaction.

|  | Partner relationship change | Family relationship change | Friends relationship change | Work colleagues relationship change |
|---|---|---|---|---|
| Face-to-face interaction change | .27* | .27* | .20* | .21* |
| Tech-mediated interaction change | −.06 | .15* | .29* | .19* |

**Notes.**
*p < .001.

Fig. 6A shows that most of the participants in a relationship that felt their relationship with their partner had improved stated their face-to-face communication had increased (78%). Whereas this percentage is substantially lower in participants reporting no change (49%) or a deterioration in relationship quality with their partner (39%). A similar pattern is also evident across the other types of relationships such as (B) family, (C) friends, and (D) co-workers, see Fig. 6. This indicates that the positive association between more face-to-face communication and improved relationship quality might be stronger than the correlation values suggest.

A similar pattern is also evident for technology-mediated communication across relationship types (except for relationship with partner). For example, the proportion of the sample that reported an improvement in friendships quality typically also reported an increased level of technology-mediated communication (85%). Whereas the amount of participants reporting an increased level of technology-mediated communication was

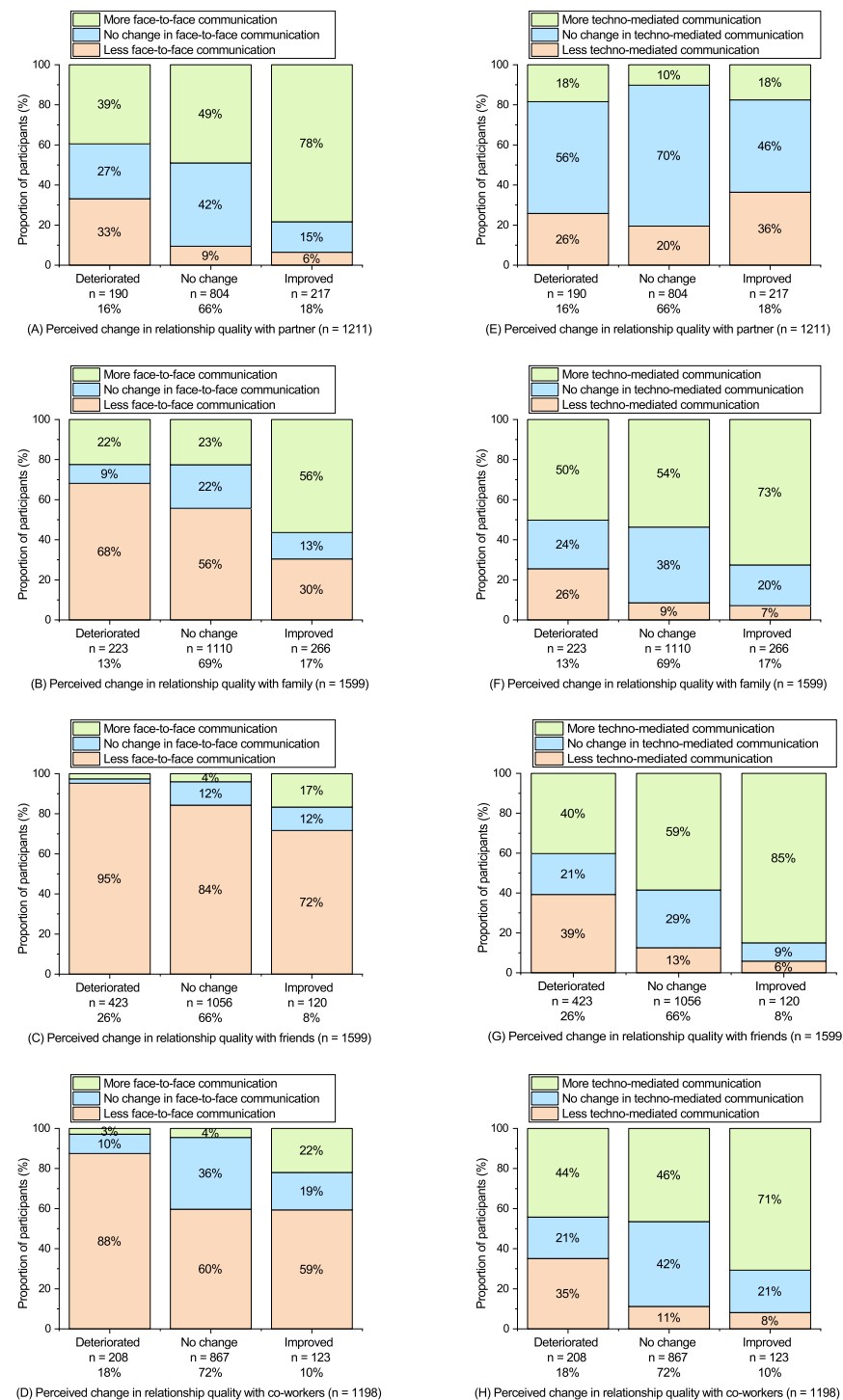

**Figure 6** **Associations between change in communication and change in perceived relationsip quality.** Charts are presented that graphically show the associations between change in communication and change in perceived relationship quality. The left column (A–D) shows change in face-to-face communication split by change in relationship quality. The right column (E–H) shows change in technology-mediated communication split by change in relationship quality. Each row presents a different relationship type 1 - Partner, 2 - Family, 3 - Friends, and 4 - Co-workers.

comparatively lower for participants reporting no change (59%) or a deterioration of friendship quality (40%), see Fig. 6G. This indicates that the positive association between more technology-mediated communication and improved relationship quality might be stronger than the correlation values suggest. Overall, our findings indicate that *both* face-to-face interaction and technology-mediated interaction had a role to play in maintaining relationships during the April lockdown period.

## DISCUSSION

The main aims of this study are to contribute to the growing body of literature that indicates government mandated lockdown policies can have potential negative consequences for public mental and physical health. We also sought to examine how changes in face-to-face and technology-mediated communication might be associated with changes in perceived relationship quality during a lockdown event. We surveyed 1599 Australian adults during a period of severe lockdown restrictions in Australia (April, 2020). Our study was conducted during a period in Australia where such a lockdown was unprecedented, and COVID-19 cases had risen dramatically just prior (i.e., in March) to the survey period. At the population level, the 5900 cases in March represented approximately 0.02% of the Australian population. At the time of the survey, the lockdown appeared to be working in curbing the spread of infections, however a great deal of uncertainty remained at the time.

From a psychological perspective, our study examines a context where the public were largely not experiencing *direct* adverse experiences from the virus (i.e., self or loved ones having the virus). Only 0.2% of our sample reported personally contracting COVID-19, 6% stated they had friends or family that had contracted COVID-19, and 10% stated they had members of their extended network with COVID-19. Therefore, we argue that in our sample participants are generally experiencing *indirect* adverse experiences via stress about the potential spread of the virus (i.e., 63% of participants reported being very/extremely concerned about their loved ones catching the virus), and adverse effects of lockdown on the economy (i.e., 73% very/extremely concerned about the economy).

### Emotional and physical well-being, exercise, concerns, social media use, and COVID-19 rumination

We found that during COVID-19 lockdown 41% and 42% of participants were experiencing greater negative than positive emotional and physical well-being, respectively. This contrasts with only 13% and 19% reporting greater negative than positive emotional and physical well-being when estimating how they were feeling the same time last year. Therefore, our results are consistent with other recent findings from around the world that the COVID-19 lockdown experience is associated with a negative impact on well-being for a substantial proportion of the public (*Biddle et al., 2020a*; *Ebrahimi, Hoffart & Johnson, 2020*; *Fisher et al., 2020*; *Gao et al., 2020*; *Huang & Zhao, 2020*; *Lee, 2020*; *Mazza et al., 2020*; *Newby et al., 2020*; *Ozamiz-Etxebarria et al., 2020*; *Phillipou et al., 2020*; *Pierce et al., 2020*; *Qiu et al., 2020*; *Roy et al., 2020*; *Twenge & Joiner, 2020*; *Wang et al., 2020*; *Westrupp et al., 2021*).

We only observed a small decrease in frequency of exercise. One potential reason for only a minor reduction in exercise is that our sample appears to have been relatively

sedentary prior to the lockdown, see Fig. 1C. Another reason is that some individuals may have increased exercise (with more time on their hands) that cancels out those exercising less. Consistent with other recent studies (*Brand, Timme & Nosrat, 2020*; *Bu et al., 2021*; *Ebrahimi, Hoffart & Johnson, 2020*; *Galle et al., 2020*; *Marashi et al., 2020*), we found a link between frequency of exercise with both physical and mental well-being suggesting that keeping physically active can potentially act as a buffer for stress during lock-down. *Codella et al. (2020)* have also pointed out that being physically active has benefits for maintaining a healthy immune system, which is obviously very relevant during a pandemic.

We acknowledge that relying on participant retrospective accounts of their mental health, physical health and exercise from the previous year is not ideal. This approach has the potential to exaggerate differences between pre- and post-lockdown ratings. This is because the reflection on last year might be biased by current feelings. However, the correlations between lockdown and last year estimates for mental health ($r = .18$) and physical health ($r = .34$) were not strong, which does provide some evidence to suggest that participants were able to dissociate current feelings from prior feelings. Comparatively, current estimate of frequency of exercise was more strongly related to prior year estimates of exercise ($r = .56$).

When asking participants to directly rate the impact they perceived COVID-19 lockdown was having on their mental health, our results are similar with another Australian survey by *Newby et al. (2020)*. In our sample 44% felt their mental health had deteriorated somewhat, and 10% a lot. In Newby et al.'s (2020) study 55% felt their mental health had worsened a little, and 23% a lot. The higher level of distress in Newby et al.'s study (2020) might be attributed to their data collection occurring slightly earlier than ours. When their study occurred, there was less clarity that the lock-down was flattening the infection rate curve. Also, they reported quite a high proportion of their participants (70%) with prior lived experience with a mental health diagnosis. We concede a limitation of our own study was that we did not query this with our own participants, however we expect that if we had such a question in our study the figure would be lower than 70%. Additionally, our sample is primarily from Western Australia whereas their sample contained more people from the Eastern states. As can be seen in Fig. 7 the number of recorded COVID-19 cases in the month of March leading up to the lockdown was higher in the Eastern states of New South Wales (NSW), Victoria (VIC) and Queensland (QLD) compared to Western Australia (WA). When considering population size, the differences were negligible, as WA has a substantially lower population size compared to these other Australian states. However, at the time the media consistently reported in terms of number of cases and communicated that the situation was worse in the Eastern states. We suggest this may have been associated with heightened anxiety in the population of the Eastern states compared to Western Australia and a potential reason for the difference between our findings with those of *Newby et al. (2020)*.

We are hesitant to frame our findings as demonstrating any substantial increase in *clinical* levels of distress. We argue that diminished well-being during such an experience (i.e., lockdown) that impacts on core psychological needs such as competence, autonomy and relatedness is a natural and understandable human reaction. We expect that most

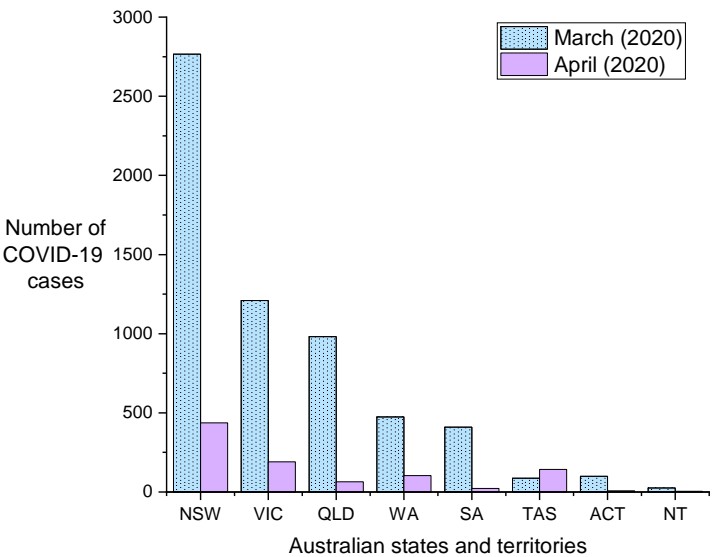

**Figure 7  COVID-19 cases in the months of March and April for each Australian state and territory in 2020.** NSW, New South Wales; VIC, Victoria; QLD, Queensland; WA, Western Australia; SA, South Australia; TAS, Tasmania; ACT, Australian Capital Territory; NT, Northern Territory. This chart was created using statistics available via the Australian government Department of Health National Notifiable Diseases Surveillance System accessible at http://www9.health.gov.au/cda/source/cda-index.cfm.

people will possess the resilience to recover quickly. However, as suggested by *Gruber et al. (2020)* there might be a proportion of participants for whom the lockdown experience acts as a traumatic precipitating event for the development of clinical depression or anxiety. Future research is required to better understand the precise extent of such occurrences.

Another finding of ours consistent with *Newby et al. (2020)* was that the dominant concerns of our participants were related to friends or family catching COVID-19 (63% very/extremely concerned), and the economy (73% very/extremely concerned). Consistent with research from other parts of the world (*Gao et al., 2020*; *Hsiang et al., 2020*; *Lee, 2020*; *Roy et al., 2020*), we found that such concern was perhaps influencing people to seek out COVID-19 information, as a large proportion of participants (i.e., 35%–55%) were monitoring sources of information about COVID-19 multiple times per day. Or alternatively, high exposure to sensationalist media coverage might have been exacerbating people's concerns (*Bendau et al., 2020*; *Bu et al., 2021*). Also, we found evidence for a high degree of rumination on COVID-19, as most participants (66%) reported thinking about COVID-19 multiple times per day. In our study we found such rumination was associated with lower levels of emotional well-being. Additionally, we found that many people (63%) reported engaging with social media more often, and consistent with *Gao et al. (2020)* we found a weak negative association with social media use and emotional well-being.

## Perceived change in social relationships and communication patterns during lockdown

In our study most participants reported no deterioration in their social relationships, and most were not concerned about it. More specifically, only 13–26% reported a deterioration
in their relationships with their partner, family, friends, or work colleagues, while 66–72% reported no change, and 8–16% reported improvement. Across the relationships of family, friends, and work colleagues, face-to-face interaction time typically decreased (54–86% of participants), while technology-mediated interaction via phone, email, or online correspondingly increased (49–57% of participants). An exception to these findings was type of communication with one's partner, with most participants responding that they had been spending more time interacting face-to-face with their partner (52%), with technology-mediated communication typically unchanged (64%).

Considering prior literature on relationship maintenance (*Blieszner & Ogletree, 2017*; *Fehr, 2004*; *Mesch & Talmund, 2006*; *Ogolsky & Bowers, 2012*; *Rossignac-Milon & Higgins, 2018*), media naturalness theory (*Kock, 2002*; *Kock, 2004*), and a general preference for face-to-face interaction (*Flaherty, Pearce & Rubin, 1998*; *Wilson et al., 2020*), at the outset of our study we were anticipating that a sudden decrease in face-to-face interaction would be associated with diminished perceived relationship quality. On the one hand, we did find that change in face-to-face interaction was positively associated with change in relationship quality. That is, participants reporting increased face-to-face interaction also tended to be more likely to report an improvement in relationship quality. This pattern was consistent across all relationship types. However, we did also find that most people reported no change in relationship quality, which indicated no systematic large scale negative impact on relationship quality during lockdown across multiple types of social relationships (i.e., partner, family, friends, and work colleagues). This finding is consistent with recent Australian research by *Biddle et al. (2020b)* whom reported that a majority of participants reported no change in relationship quality with other people/family members in their household.

We offer a couple of complimentary potential explanations for this finding. First, while lockdown drastically reduces opportunity for many types of relationship maintenance experiences (e.g., going to see a movie at a cinema), it arguably also provides an over-arching experience that has potential to be a source of relationship bonding. In our study, 94% of participants reported 'talking with people about COVID-19' at least a few times per week, and 45% reported multiple times per day. Additionally, the pandemic appears to be a largely negative experience for many people around the world (*Benke et al., 2020*; *Ebrahimi, Hoffart & Johnson, 2020*; *Fiorenzato et al., 2020*; *Gao et al., 2020*; *Hamadani et al., 2020*; *Holman et al., 2020*; *Huang & Zhao, 2020*; *Kalaitzaki, 2020*; *Lee, 2020*; *Marashi et al., 2020*; *Mazza et al., 2020*; *Ozamiz-Etxebarria et al., 2020*; *Pierce et al., 2020*; *Qiu et al., 2020*; *Roy et al., 2020*; *Twenge & Joiner, 2020*; *Wang et al., 2020*; *Zacher & Rudolph, 2020*), and we have reported a substantial proportion in our sample of Australians being negatively affected during lockdown. Numerous studies have indicated that shared experiences of pain, adversity, and/or hardship have the potential to act as social bonding experiences (*some examples*: *Bastian, Jetten & Ferris, 2014*; *Bastian et al., 2014*; *Bastian et al., 2018*; *Breslin, 2019*; *Shaw, Pollio & North, 2020*).

The second potential reason why we did not find any substantial negative impact on social relationships is that we found many participants were making good use of communications technology to stay connected during the lockdown period. Across

relationship types of family, friends, and work colleagues there was a positive association with change in technology-mediated communication and change in relationship quality. That is, participants reporting an increase in their technology-mediated communication were more likely to also report an improvement in relationship quality. Therefore, while modern communication technology such as social media has potential to reduce well-being via increasing rumination (*Ohannessian, Fagle & Salafia, 2020*; *Parris et al., 2020*) and excessive social comparison (*Feinstein et al., 2013*), such technologies are also arguably particularly useful for helping people keep informed and connected with others during lockdown experiences.

## Study limitations

Our sample primarily consists of females (78%), in a relationship (75%), living in Western Australia (81%). Therefore, our sample is not representative of the entire Australian population so generalisations from our findings should be made with caution. Our survey was advertised primarily on Facebook about people's experience during COVID-19 which may have attracted respondents who were concerned and stressed about COVID-19 to participate. Therefore, our results indicating an overall decrease in emotional/physical well-being, and high concerns about loved ones contracting the virus and concern for the economy, may be over-represented in our specific sample. Also, our sample is limited to people who are actively engaged with social media. Therefore, our findings regarding the extent of social media use, and technology-mediated communication use more broadly, may also be over-represented in our specific sample.

In our study participants had been under COVID-19 lockdown for 2-4 weeks. It could be that a more protracted lockdown would produce quite a different result. While technology-mediated communication appeared to compensate for reduced face-to-face time in the present study, in a lengthier lockdown situation there remains the distinct possibility that reduced face-to-face contact could result in degraded social relationships. There is a need for the further development of communication technologies to be better prepared for such future possibilities. An area of current development is communication in virtual reality via head mounted display (HMD) (*Pan & Hamilton, 2018*; *Rosedale, 2016*; *Seymour, Reimer & Kay, 2018*; *Seymour et al., 2019*). HMD-based virtual reality has the potential to become a technology-mediated mode of communication that can psychologically feel near identical to face-to-face interaction (*Pan & Hamilton, 2018*; *Rosedale, 2016*; *Seymour, Reimer & Kay, 2018*). Future research is needed to fast-track the development of this new mode of social interaction.

With a cross-sectional design, we are unable to make any confident conclusions regarding the direction of causality for the associations between variables in our study. For example, we have suggested that exercise might be a protective factor against stress, however it may also be that stress reduces motivation for exercise (*Stults-Kolehmainen & Sinha, 2014*). We have suggested that engaging in greater social media use might be exacerbating stress, but it could also be that stress is what is driving people to seek out more social connection on social media (*Luo & Hancock, 2020*). We have suggested that engaging in greater social interaction face-to-face and via technology-mediated modes of communication can help

to decrease stress, but again it could be that it is stress that is motivating people to increase their interaction with others as they seek social support (*Sameer et al., 2020*).

## CONCLUSIONS

Our study within an Australian sample is consistent with emerging literature around the world that lockdown experiences during COVID-19 are associated with decreased emotional and physical well-being for a substantial proportion of individuals. The highest concerns reported by our participants were about themselves and loved ones catching the virus, and concern for the economic impact. Many participants were found to be engaging heavily with the media to stay informed, leaving them prone to rumination about COVID-19 that for some appeared to be exacerbating their stress. We found evidence to suggest that despite such stressors, people were using technology to stay connected with others so that overall, there was not any major impact on social relationships for most individuals in our study. An increase in social media use was however also found to be a factor that might exacerbate stress, while staying physically active a potential buffer against stress.

### Funding
The authors received no funding for this work.

### Competing Interests
The authors declare there are no competing interests.

### Author Contributions
- Shane L. Rogers conceived and designed the experiments, performed the experiments, analyzed the data, prepared figures and/or tables, authored or reviewed drafts of the paper, and approved the final draft.
- Travis Cruickshank conceived and designed the experiments, performed the experiments, authored or reviewed drafts of the paper, and approved the final draft.

### Human Ethics
The following information was supplied relating to ethical approvals (i.e., approving body and any reference numbers):

Edith Cowan University granted ethical approval for this research (Ethical Application Ref: 2020-01305-ROGERS).

### Data Availability
Raw data is available in the Supplemental Files.

### Supplemental Information
Supplemental information for this article can be found online at http://dx.doi.org/10.7717/peerj.11767#supplemental-information.

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
