# Peer review of "Change in mental health, physical health, and social relationships during highly restrictive lockdown in the COVID-19 pandemic: evidence from Australia"

_PeerJ, doi:10.7717/peerj.11767_

## Round 0.1 · original submission · Major Revisions

Thank you for your submission. The reviewers have identified a number of areas to be addressed in your re-submission.

·

Basic reporting

The basic reporting and the literature context is good. But it is tad long and mostly redundant. Usually Introductions are kept short and usually of three paragraphs... without any figures or Table to refer to.

Some contextual citations like sameer et al., 2020 is missing.

First paragraph of discussion is redundant. There are many portions of the paper which have been repeated either in introduction, results or discussion. .

Experimental design

It is well designed and implemented. The explanations are long and repetitive. It5 would be better to use figure an tables as supportive things to carry the message, rather to give description of everything..

Validity of the findings

The findings are robust and meaningful

Additional comments

Dear Authors,
The paper is well envisaged and robust in message. But is a bit tad long. And there are many redundancies in it.
Some of the papers or covid like Sameer et al., 2020 have not been cited.
I would recommend to shorten it and reduce the overlap of in the different sections of the paper.
Regards

Reviewer 2 ·

Basic reporting

The author might avoid frequent compound sentences for clarity and easy readability.

Experimental design

Nil

Validity of the findings

Nil

Additional comments

The manuscript titled “Change in mental health, physical health, and social relationships during highly restrictive lockdown in the COVID-19 pandemic: Evidence from Australia" addresses an important public health issue which is highly relevant in the current global scenario.
Though the article has been written well in general, the author may address the following issues to improve the manuscript further.
1. Introduction is lengthy. The author could make it precise and concise narrowing down to the research problem.
2. Materials and methods: Under "Perceived impact of COVID19 lockdown on different aspects of life, and concerns about COVID19"- Interpretation of the overall score for impacts and concerns should be mentioned clearly for better understanding since each have different interpretation.
3. Results: Line 355, change "increases to increased"
Line 357, the cited figure number doesn't match with the results explained. (Figure 2 instead of Figure 4?)
Interpretation of the effect sizes should be reconsidered.
Line 396, spearman r value mentioned (0.63) and the value mentioned in the table (0.64) does not match
Line 394 and 395 - change participant to participants.
In general, the author may verify the values mentioned in the manuscript matches that with the tables and figures (rounding off the numbers)
Line 463, cite table 3 also
Line 466, Replace figure 7A with 7C
Line 466- 470, needs more clarity and may be broken down into simple sentences
Line 471, the author states "similar pattern is also evident". However the result interpretation was different for friends and coworkers. This statement needs more clarity.
Line 474, mentions "8% of the sample" which does not match the value given in the figure (6 %). Similar discrepancies in line 476, 478 and 479.
4. DIscussion: Line 495, replace increase with spread
Line 497, "our study examines a context where the public were largely not
498 experiencing direct adverse experiences from the virus". The authors may specify the % of participants who were COVID positive and those who recovered instead of "largely not".
Line 515, add the % of frequency of exercise to improve clarity.
Line 515, specify whether the author refers to emotional or physical wellbeing or both
Line 537, "Also, they reported quite a high proportion of their participants (70%) with prior lived
experience with a mental health diagnosis". Why was the prior history of mental illness not taken into account in the present study. Justify it. Is this another limitation of this study?
5. The conclusion sections appears too lengthy and repetitive and includes points that can be moved to the discussion.

---

## Round 0.2 · Minor Revisions

Thank you for your re-submission. The manuscript is much improved, however there are some minor revisions required by the reviewers.

·

Basic reporting

The authors have modified the manuscript as per the suggestions.

Experimental design

NA

Validity of the findings

NA

Additional comments

NA

Reviewer 2 ·

Basic reporting

Thanks for the author for the substantial improvement in the manuscript.
However, I would like to point out a few points
1. Abstract section doesnt have conclusion. It has be clubbed together with results section
2. Though the author has made a significant effort in reducing the length of manuscript, the introduction section needs to be curtailed down considerably as most portions are still redundant. The authors could narrow down to the research problem and give background pertaining to those in a crisp way.
3. Limitations section needs to be curtailed

Experimental design

Nil

Validity of the findings

Nil

Additional comments

In general, the manuscript is way to lengthy for an original article. As pointed out by earlier reviews too, introduction and limitation section needs to be curtailed as its way too lengthy.

---

## Round 0.3 · accepted · Accept

Thank you for your re-submission. I am pleased to tell you this has now been accepted for publication!